# Land-Use Type Drives Soil Population Structures of the Entomopathogenic Fungal Genus *Metarhizium*

**DOI:** 10.3390/microorganisms9071380

**Published:** 2021-06-25

**Authors:** María Fernández-Bravo, Florian Gschwend, Johanna Mayerhofer, Anna Hug, Franco Widmer, Jürg Enkerli

**Affiliations:** 1Molecular Ecology, Agroscope, CH-8046 Zürich, Switzerland; mariadelcarmen.fernandezbravo@agroscope.admin.ch (M.F.-B.); florian.gschwend@agroscope.admin.ch (F.G.); johanna.mayerhofer@agroscope.admin.ch (J.M.); franco.widmer@agroscope.admin.ch (F.W.); 2Swiss Soil Monitoring Network (NABO), Agroscope, CH-8046 Zürich, Switzerland; anna.hug@agroscope.admin.ch

**Keywords:** *M. brunneum*, *M. robertsii*, *M. guizhouense*, microsatellite, SSR, EF-1alpha, abiotic factors, arable land, grassland, forest, biological control

## Abstract

Species of the fungal genus *Metarhizium* are globally distributed pathogens of arthropods, and a number of biological control products based on these fungi have been commercialized to control a variety of pest arthropods. In this study, we investigate the abundance and population structure of *Metarhizium* spp. in three land-use types—arable land, grassland, and forest—to provide detailed information on habitat selection and the factors that drive the occurrence and abundance of *Metarhizium* spp. in soil. At 10 sites of each land-use type, which are all part of the Swiss national soil-monitoring network (NABO), *Metarhizium* spp. were present at 8, 10, and 4 sites, respectively. On average, *Metarhizium* spp. were most abundant in grassland, followed by forest and then arable land; 349 *Metarhizium* isolates were collected from the 30 sites, and sequence analyses of the nuclear translation elongation factor 1α gene, as well as microsatellite-based genotyping, revealed the presence of 13 *Metarhizium brunneum*, 6 *Metarhizium robertsii*, and 3 *Metarhizium guizhouense* multilocus genotypes (MLGs). With 259 isolates, *M. brunneum* was the most abundant species, and significant differences were detected in population structures between forested and unforested sites. Among 15 environmental factors assessed, C:N ratio, basal respiration, total carbon, organic carbon, and bulk density significantly explained the variation among the *M. brunneum* populations. The information gained in this study will support the selection of best-adapted isolates as biological control agents and will provide additional criteria for the adaptation or development of new pest control strategies.

## 1. Introduction

The genus *Metarhizium* Sorokīn (Hypocreales: Clavicipitaceae) includes more than 30 described species (including both asexual and sexual states) that are pathogenic to arthropods, particularly insects and some arachnids [1,2]. The wide host spectrum of *Metarhizium* spp. includes many important crop pests such as *Helicoverpa armigera* (Hübner, 1805), *Diabrotica virgifera virgifera* (LeConte, 1868), *Agriotes* spp. (Eschscholtz, 1829), and also termites, cockroaches, and even disease-transmitting insects such as tse-tse flies [2,3,4,5,6,7]. Based on the pathogenicity of this fungus, a number of biological control agents have been developed as commercial mycoinsecticides and mycoacaricides to control pest arthropods worldwide (see [8] and https://www.eppo.int; https://www.epa.gov).

Soil is considered the main habitat and reservoir of *Metarhizium* spp., and they have been isolated from very different ecosystems, from tropical to temperate, semiarid, and artic areas [5,9]. Besides being entomopathogenic, *Metarhizium* spp. exhibit a pronounced saprophytic lifestyle, and, for some isolates and/or species, symbiotic interactions with plants as rhizosphere colonizers and/or endophytes have been reported [9,10]. Studies have demonstrated, for instance, that isolates of *M. robertsii* J.F. Bisch., S.A. Rehner & Humber, *M. brunneum* Petch, and *M. guizhouense* Q.T. Chen & H.L. Guo can provide insect-derived nitrogen to plants [11,12], increase plant growth and productivity [13,14], as well as protect plants from environmental stresses, for example, in saline environments [15,16]. The versatile roles *Metarhizium* spp. may have emphasize the beneficial potential and value of its presence, particularly in agricultural habitats [17].

Arable land, grassland, and forests represent major anthropogenic land-use types (LUTs) and cover about 40% of the global land area [18]. A number of studies have reported possible effects of LUTs on the presence and abundance of *Metarhizium* spp. Factores responsible includ crop types, farming practices management intensity, tillage, and the application of pesticides or fertilizers [19,20,21,22]. For instance, it has been consistently reported that *Metarhizium* spp. occur at higher abundances in unforested, sun-exposed LUTs or open land areas such as arable- and grasslands compared to forests, where their abundances are lower or sometimes not even detected [17,19,23,24,25,26]. The abundance of *Metarhizium* spp. has been more prominent in unforested semi-natural habitats such as permanent grassland and field margins compared to arable land in Switzerland [27,28]. Furthermore, there is evidence from field studies in Norway indicating a higher abundance of entomopathogenic fungi such as *M. anisopliae*, *Beauveria bassiana* (Bals.-Criv.) Vuill., or *Tolypocladium cylindrosporum* W. Gams in soils of organically managed fields compared to conventionally managed fields [29]. Several studies have suggested that the application of organic fertilizers may be the main driving factor for the increased abundance of *Metarhizium* spp. [22,30].

While the overall abundance of *Metarhizium* spp. in different LUTs has been explored in various studies, information on the species composition of *Metarhizium* communities and within-species diversity in different LUTs is limited. In addition, many of these studies have been performed prior to the establishment of the current *Metarhizium* species concept, which has precluded a comparison of data from these different periods. Nevertheless, the results reported by Cabrera-Mora et al. [31] have indicated that cropping systems can affect species abundances, as *M. brunneum* was isolated predominantly from soil where beans were grown and *M. robertsii* from soils where maize was grown. Similarly, Wyrebek et al. [32] reported that *M. robertsii*, *M. brunneum*, and *M. guizhouense* were closely associated with soils from grassland, shrubs, or trees, respectively. In a study performed in Ontario, Canada, two genetically distinct groups were found to be associated with agricultural unforested field habitats and forested soils, respectively [23,33,34]. Isolates of the two genetic groups were subsequently described as *M. brunneum* and *M. robertsii* [35]. Studies assessing within-species diversity have mostly focused on individual fields or agricultural systems only, e.g., an agricultural field in Denmark [36] and a strawberry field in Brazil [37]. A comparison of the within-species diversity of *M. brunneum* and *M. robertii* between agricultural and adjacent natural habitats has been performed in different ecological regions in northwestern North America [38]. While considerable genotype diversity has been reported for both species in both habitats, no significant differences have been detected between them. Information on the number of species, their population structures, and distributions in different LUTs remains limited. This information, however, is important to understand the interaction of ecological conditions and *Metarhizium* population structures. In turn, it may allow the improvement of existing habitats or the development of new habitats and/or LUT-adapted biological control approaches, including conservation biological control [17].

Conditions in a particular soil or at a particular site are affected, on the one hand, by anthropogenic activities and, on the other hand, by different environmental factors, i.e., abiotic (physical and chemical), biotic (derived from organisms), as well as climatic (humidity, temperature, and solar radiation) factors. All these factors may have direct effects on the occurrence, abundance, propagation, persistence, and even virulence of *Metarhizium* populations [24,39,40,41,42]. Soil physical, chemical, and biological factors, such as soil texture, pH, and organic matter content, belong to the most intensively studied factors in this context. However, various reports have not been able to provide consistent information, and, in many cases, they have been contradictory. For example, organic matter, which represents an important source of organic carbon in soil, has been reported to negatively correlate with the abundance of *Metarhizium* in agricultural land in Pennsylvania, USA [20], whereas other studies have reported positive correlations to agricultural land, grassland, and forests in Spain [24]. Similarly, the C:N ratio, a soil factor directly related to the quality of organic matter, has been reported to positively correlate with the abundance of entomopathogenic fungi in soil [43]. In that study, the C:N ratio was proposed as a predictor of *Metarhizium* spp. abundance in vineyard soils; however, whether this concept may generally be applicable to vineyards or even other habitats or LUTs such as grassland and arable land or even forests is currently not clear. The effect of soil pH on microbial populations is well documented; however, data on the specific effects of pH on *Metarhizium* abundance are currently not conclusive. While some studies suggest that *Metarhizium* prefer more alkaline environments, others have reported that *Metarhizium* are more adapted to acidic conditions or are not even responsive to pH [24,44].

In the present study, we determine *Metarhizium* spp. abundance and population structure at 30 different monitoring sites of the Swiss soil-monitoring network (NABO) that equally represent arable land, grassland, and forest. The overall goal was to obtain profound information on habitat selectivity and the factors that may drive population structures and the abundance of this important entomopathogenic fungal genus. Our specific goals were to: (1) determine *Metarhizium* spp. abundance as well as species diversity and population structures at the 30 sites across Switzerland, (2) investigate the differences among and within LUTs in detected *Metarhizium* populations, and (3) assess the extent to which soil physical, chemical, and biological factors may explain the differences observed.

## 2. Materials and Methods

### 2.1. Sampling Sites, Sampling Procedure

The Swiss soil-monitoring network (NABO, www.nabo.ch), established in 1984, monitors 103 sites for physical, chemical, and biological parameters (biomass, soil respiration) [45]. The sites represent three different LUTs, i.e., arable land, permanent grassland, and forest, which dominate and characterize the landscape in the midlands and pre-alpine areas in Switzerland. In 2012, the NABO was complemented with the NABObio initiative to monitor soil microbial biodiversity at 30 NABO sites (10 sites of each of the three LUTs) (Appendix A) [46].

Sampling was carried out in spring after snowmelt and before the first fertilization, starting in Ticino (south of Switzerland) at the beginning of February 2016 and finishing in Grindelwald (alpine area) at the end of June 2016. This sampling scheme was implemented to harmonize time points in relation to the vegetation status of individual sites, i.e., the beginning of vegetation growth. Three composite samples were collected at each of the 30 sites. Composite samples consisted of 25 bulked soil cores (2.5 cm diameter × 20 cm depth) that were taken evenly distributed in a georeferenced 10 × 10 m plot, according to the standardized sampling protocol of NABO sampling [47]. In total, 90 composite samples were collected (Appendix A) and transported to the laboratory immediately after sampling and kept at 4 °C until use. Thirteen soil factors (physical, chemical, and biological), for example, C:N ratio (organic carbon/total nitrogen), bulk density, and basal respiration, among others, were measured from each sample, as described previously by Gubler et al. [45]. In addition, mean annual temperature (MAT) and mean annual precipitation (MAP) were determined based on yearly data from 1981 to 2015 from MeteoSwiss (http://www.meteoswiss.admin.ch). In total, 15 environmental factors were evaluated in this study (Appendix A).

### 2.2. Fungal Isolation and DNA Extraction

*Metarhizium* abundance was determined, and single colonies were isolated from each of the three mixed soil samples of the 30 sites. The samples were sieved with a 2 mm mesh, and soil water content was determined. Then, 5 g of each homogenized sample was suspended in 25 mL of sterile distilled water + 0.1% Tween 80 in a 100 mL Erlenmeyer flask. Flasks were stirred in a rotatory shaker at 120 rpm at room temperature for 3 h. After 20 s of sedimentation, 100 µL aliquots of the suspension were spread on 90 mm Petri dishes containing a semi-selective medium (SM) with dodine (Discovery, Leu Gygax AG, Switzerland) as the selective compound [48]. Two replicates were plated per soil sample. Cultures were grown for 15 d at 24 ± 1 °C in the dark and inspected every 2 days. The fungal isolates were classified to the genus level using taxonomic keys [49]. *Metarhizium* colony forming units (CFU) per g (dry weight) of soil were determined for each plate and means calculated per site. Three sporulating colonies were randomly taken per plate. Conidia of each colony were plated with a dilution smear in a new SM plate, and a monosporic colony was isolated. In total, a maximum of 18 isolates was collected per site. All the fungal isolates obtained were deposited in the culture collection of the Molecular Ecology Group, Agroscope, Zürich, Switzerland.

Monosporic cultures were plated onto sterile filter paper placed on potato dextrose agar (PDA) plates to produce mycelium for DNA extraction. After 4 days of incubation at 24 ± 1 °C, mycelium was scraped off the filter paper with a sterile scalpel, placed in an Eppendorf tube, and lyophilized for six hours at −4 °C using a CentriVap benchtop centrifugal vacuum concentrator (LabConco, Kansas City, MO, USA). Genomic DNA of each isolate was extracted using the NucleopSpin Plant II kit (Machery & Nagel, Düren, Germany). DNA concentration was determined using a Cary Elipse fluorescence spectrophotometer (Varian, Palo Alto, CA, USA) with a Picogreen^®^ fluorescent nucleic acid stain (Invitrogen, Carlsbad, CA, USA). The DNA extracts were standardized to 5 ng µL^−1^.

### 2.3. Sequence Analysis

*Metarhizium* isolates were taxonomically identified to the species level by sequencing the 5′ end of the nuclear translation elongation factor-1α (5′-TEF-1α) and subsequent alignment with reference sequences. The 5′-TEF region was amplified using primers EF1T 5′-ATGGGTAAGGARGACAAGAC-3′ [50] and EFjmetaR 5′-TGCTCACGRGTCTGGCCATCCTT-3′ [51]. PCRs were performed in 20 µl reaction volumes consisting of 15 ng genomic DNA and 1x Phusion HF buffer containing 7.5 mM MgCl_2_, 0.2 mM dNTPs, 0.2 µM of each primer, 3% dimethyl sulfoxide (DMSO), and 0.2 U of Phusion Hot Start II High-Fidelity DNA polymerase (Thermo Fisher Scientific, Waltham, MA, USA). PCR conditions were: 30 s denaturation at 98 °C, followed by 38 cycles of 5 s denaturation at 98 °C, 20 s annealing at 58 °C, and a 1 min extension at 72 °C. Reactions were completed with a 10 min elongation at 72 °C.

PCR products were cleaned up using the NucleoSpin^®^ Gel and PCR Clean-up kit (Machery & Nagel, Düren, Germany), following the manufacturer’s protocol. Sequencing reactions were performed with primers EF1T and EFjmetaR using the BigDye™ Terminator v3.1 Cycle Sequencing kit (Applied Biosystems, Foster City, CA, USA) and sequences determined using an ABI PRISM 3130xl genetic analyzer, as described above. Sequences were assembled and corrected using the software DNABaser V.4 (Heracle Biosoft, Pitești, Romania). The obtained sequences were aligned with reference sequences of *Metarhizium* spp. type isolates for species allocation, as described in Mayerhofer et al. [51]. The sequences of one isolate per multilocus genotype (MLG) are deposited at GeneBank (accession numbers MZ297396 to MZ297417, Appendix B). Reference sequences were downloaded from GenBank, and alignments were performed using the Clustal-W subsequence realignment tool implemented in MEGA X [52]. The maximum likelihood method based on the Kimura 2-parameter model implemented in MEGA X was used to calculate (1000 iterations) and construct a phylogenetic tree [53].

### 2.4. Multilocus Microsatellite Genotyping

Fourteen microsatellite markers were used in five primer pair sets, as described by Mayerhofer et al. [54], to assess the genotype of the collected *Metarhizium* isolates. Multiplex PCRs were performed in 20 µL reaction volumes containing 5 ng of genomic DNA, 1× GoTaq colorless Flexi Buffer, 0.2 mM of each primer (forward primer labeled with FAM, HEX, or NED), 0.2 mM of dNTPs, 3 or 4 mM of MgCl2 (depending on the multiplex set, [54]) and 0.25 U of GoTaq G2 Flexi DNA Polymerase (Promega, Madison, WI, USA). Cycling conditions included an initial denaturing step of 2 min at 94 °C, followed by 12 touchdown cycles of 30 s denaturation at 94 °C, 30 s annealing at Ta + 12 °C, (with a 1 °C decrease per cycle), and 40 s of extension at 72 °C. This was followed by another 22 or 30 cycles [54] of 30 s denaturation at 94 °C, 30 s at Ta [54], and a 40 s extension at 72 °C. Reactions were completed with a 15 min elongation at 72 °C. PCR product sizes (microsatellite allele size) were determined on an ABI 3500 Series genetic analyzer (24 capillaries) using POP-7 polymer (Applied Biosystems, Foster City, CA, USA). GenScan ROX400 (Applied Biosystems) was used as an internal size standard. Data were analyzed using GenMarker V2.4.0 (SoftGenetics, State College, PA, USA) and allele sizes corrected according to fragment sizes of reference strains *M. brunneum* ARSEF7524 and *M. robertsii* ARSEF7532.

### 2.5. Data Analysis

Analysis of variance (ANOVA) was used to assess the effect of LUT, site, and soil environmental factors on the abundance of *Metarhizium* (CFU g^−1^ of soil dry weight), followed by the Tukey-HSD test to assess pairwise differences. Sites with zero *Metarhizium* spp. abundance were not included in the calculations of differences among LUTs to provide a clear representation of mean fungal abundance where *Metarhizium* was detected.

Microsatellite marker data analyses were performed with the R package Poppr [55]. Data were clone-corrected, and the number of MLGs calculated. The Shannon-Wiener index (H), [56] and the evenness index (E.5) [57,58,59] were calculated for each *Metarhizium* species and LUT based on the occurrence of each MLG.

Differences in the population structure of *Metarhizium* MLGs among LUTs and species were assessed with overall and pairwise PERMANOVA based on Bray–Curtis (BC) dissimilarity matrices [60] using the function “adonis” within the vegan R package, “pairwise.perm.manova” within the RVAideMemoire R package [61,62], and the Benjamini–Hochberg *p*-value correction in R [63]. Effects of soil factors on *Metarhizium* populations were assessed with overall PERMANOVA tests for each factor individually using the “adonis” function. Principal coordinate analyses (PCoA) were performed based on BC dissimilarity matrices using the “cmdscale” function included in the R core package [63,64]. The “envfit” function from the vegan R package was used to plot the correlations between *Metarhizium* species population structures and the significant soil factors determined in a PCoA ordination.

## 3. Results

### 3.1. Abundance of Metarhizium in Soil

Using a semi-selective medium, the genus *Metarhizium* was detected at eight of ten arable land sites, at all ten permanent grassland sites, and at four of ten forest sites (Figure 1 and Appendix A). LUT significantly (F_2,19_ = 4.30; *p*-value = 0.0287) affected the abundance of *Metarhizium*, determined as CFU g^−1^ soil dry weight. Pairwise tests (Tukey-HSD test, *p*-value < 0.05) revealed significant differences between the site means of grassland (3501.2 ± 2254.2 CFU g^−1^ soil dry weight) and arable land (1199.0 ± 1278.5 CFU g^−1^ soil dry weight) but not between the means of grassland and forest (1620.8 ± 938.2 CFU g^−1^ soil dry weight) or arable land and forest. Significant differences (F_29,58_ = 11.63; *p*-value < 0.0001) in *Metarhizium* abundances were also detected among sites (Figure 1 and Appendix A), where abundance ranged from 0 to 6362.5 ± 1467.2 CFU g^−1^ soil dry weight (mean ± SD).

*Metarhizium* abundances at the 30 sites were grouped into three distinct groups, i.e., sites with low abundance (mean abundance < 150 CFU g^−1^ soil dry weight), medium abundance (mean abundance ranging between 150 and 4000 CFU g^−1^ soil dry weight), and high abundance (sites with a mean abundance > 4000 CFU g^−1^ soil dry weight) (Figure 1 and Appendix A). Medium *Metarhizium* abundance was detected at sites of all three LUTs (14 sites), whereas high abundance was only detected in grassland (5 sites). The low abundance group included sites where *Metarhizium* was detected with a low abundance (3 sites in arable land) as well as sites where *Metarhizium* was not detected (2 sites in arable land, and 6 sites in forest). Arable land was the only LUT that included sites with low abundance as well as sites where the fungus was not detected (5 sites). However, due to missing statistical power, no discrimination of these sites was possible, and they were all combined in the group of low abundance.

### 3.2. Effect of Environmental Factors on Metarhizium Abundance Groups

In total, 15 environmental factors were measured at the 30 sites and assessed for differences among the sites categorized according to the three *Metarhizium* abundance groups (Table 1 and Appendix A). Two factors, the C:N ratio and mean annual precipitation (MAP), differed significantly among the *Metarhizium* abundance groups independent of the three LUTs between high and low as well as medium and low abundance groups but not between high and medium abundance groups (Table 1 and Appendix A). Within each LUT, different factors were significant between the low, medium, and high abundance groups. In arable land, the six soil factors—altitude, clay, sand, soil skeleton, total carbon, and DNA (proxy for biomass)—differed significantly between the low and medium abundance groups (Table 1 and Appendix A). In grassland, only silt significantly differed between high and medium abundance, whereas in forest, altitude, clay, sand, and C:N ratio significantly differed between medium and low *Metarhizium* abundance (Table 1, Appendix A).

### 3.3. Metarhizium Species Occurrence and Genetic Diversity

In total, 349 isolates were collected from the 22 sites at which *Metarhizium* was detected and, based on sequence analyses of the 5′ end of nuclear translation elongation factor-1α (5′-TEF-1α), assigned to three *Metarhizium* species: *M. brunneum* (259 isolates), *M. robertsii* (80 isolates), and *M. guizhouense* (10 isolates) (Table 2 and Appendix A). *M. brunneum* was present in all three LUTs (7 arable land sites, 10 grassland sites, 4 forest sites), whereas *M. robertsii* and *M. guizhouense* were present in arable land (*M. robertsii* 6 sites, *M. guizhouense* 1 site) and grassland only (*M. robertsii* 4 sites, *M. guizhouense* 2 sites; Table 2 and Figure 2). The proportion of *M. brunneum* and *M. robertsii* isolates was similar in the low (61% and 39%) and medium (56% and 40%) abundance groups in arable land and the high (57% and 36%) abundance group in grassland (Figure 2). In contrast, in the medium abundance group in grassland, the portion of *M. brunneum* isolates was higher, i.e., *M. brunneum* 94% and *M. robertsii* 5% (Figure 2). *M. guizhouense* was detected in the medium abundance group in arable land (4%) and grassland (1%) and the high abundance group in grassland (7%) (Figure 2).

Genotyping of the isolates based on microsatellite analyses revealed 22 different MLGs among the 349 isolates: 13 MLGs for *M. brunneum*, 6 MLGs for *M. robertsii*, and 3 MLGs for *M. guizhouense*, revealing that these microsatellite-based MLGs were species-specific (Figure 3 and Table 2). Overall, the number of detected MLGs was similar in arable land and grassland, while it was 2.5 to 3 times lower in forest. For *M. brunneum*, the number of MLGs was higher in grassland compared to arable land and forest, whereas for *M robertsii*, the number of MLGs was higher in arable land. Shannon–Wiener genetic diversity (H) for *M. brunneum* was highest in permanent grassland, followed by arable land and forest, while for *M. robertsii*, diversity was higher in arable land and lower in grassland (Table 2). In contrast, the evenness (E.5) of *M. brunneum* was higher in forest compared to grassland and arable land, and the evenness of *M. robertsii* was higher in grassland compared to arable land (Table 2). Due to the low number of *M. guizhouense* isolates identified, diversity indices for this species were not calculated.

In total, 18 of the 22 MLGs were specific to one of the three LUTs. Six MLGs were specific to arable land (two *M. brunneum*, three *M. robertsii*, one *M. guizhouense*), eight to permanent grassland (five *M. brunneum*, one *M. robertsii*, two *M. guizhouense*), and four to forest, which clustered separately within *M. brunneum* in the 5′-TEF-1α-based phylogenetic tree (Figure 3). None of the MLGs were detected in all three LUTs, while the four most abundant MLGs, i.e., two *M. brunneum* (15 and 19) and two *M. robertsii* MLGs (2 and 13), were present in arable land and grassland and represented 63.61% of all *Metarhizium* isolates collected (Figure 3). The four forest-specific MLGs represented 18.05% of all *Metarhizium* spp. isolates investigated. The abundance of individual MLGs did not correlate with affiliation to particular *Metarhizium* abundance groups, e.g., *M. brunneum* MLG15 and *M. robertsii* MLG 13 were present in the low, medium, and high abundance groups in different sites (data not shown). Eleven MLGs (five *M. brunneum*, three *M. robertsii*, and three *M. guizhouense*) were isolated from single sites only (Figure 3). In contrast, MLG 15 comprised isolates from 15 different sites. Two MLGs (MLG 4 and 5) of the four forest-specific MLGs included isolates from all the four sites and two included isolates from two sites (MLG 6) and one (MLG 10) site where *M. brunneum* was detected (Figure 3).

### 3.4. Land-Use Type Effect on Metarhizium Populations

Overall, PERMANOVAs indicated that the three LUTs significantly explained the variation in *M. brunneum* population structures (R^2^ = 0.37; pseudo-F = 5.36; *p*-value = 0.0001) (Table 3). PCoA explained 47.00% of the observed variation among *M. brunneum* populations, i.e., 24.40% of the variation in the first axis, separating forest populations from arable land and grassland populations and 22.60% of the variation in the second axis, differentiating between arable land and grassland populations (Figure 4A). Pairwise PERMANOVA tests among the three LUTs revealed significant differences between *M. brunneum* populations from forest and arable land (*p*-value = 0.003), between forest and grassland (*p*-value = 0.003), but not between grassland and arable land (*p*-value = 0.05). Overall, PERMANOVA indicated that the LUT (arable land, grassland) did not significantly explain the variation among *M. robertsii* population structures (*p*-value = 0.466) (Table 3). PCoA explained 70.40% of the observed variation among *M. robertsii* arable land and grassland populations, i.e., 52.00% of the variation in the first axis and 28.40% of the variation in the second axis (Figure 4B). Variations among *M. guizhouense* populations were not evaluated further due to the low number of isolates (10) and genotypes (3 MLGs) detected.

### 3.5. Effect of Environmental Factors on Metarhizium Populations

PERMANOVA revealed that 5 of the 15 environmental factors significantly explained the variation of *M. brunneum* population structures among the three LUTs (Table 3 and Figure 4A). Most variation was explained by the factor C:N ratio (25%), whereas basal respiration, total carbon, organic carbon, and bulk density explained the variation in the same range of 11% to 13%. Soil skeleton was the only factor that significantly explained the variation among *M. robertsii* populations from arable land and grassland (Table 3 and Figure 4B). Except for the factor C:N ratio, which significantly explained the variation of *M. brunneum* populations among grassland sites, no other environmental factor significantly explained the variation of *M. brunneum* or *M. robertsii* populations within arable land, grassland, or forest.

## 4. Discussion

Identification of the factors that drive population structure and development of insect pathogenic fungi such as *Metarhizium* spp. is a fundamental requirement to exploit their potential in biological control. The results presented in this study emphasize the effect and influence of particular LUTs on *Metarhizium* abundance and population structures and indicate the complexity of the factors involved. Populations at the 30 sites investigated were composed of the three species *M. brunneum*, *M. robertsii*, and *M. guizhouense*, of which *M. brunneum* (with 74% of isolates recovered) was the most abundant and genotypically diverse species and *M. guizhouense* (with 3%) the least abundant of all isolates collected. Distinct differences in *Metarhizium* population structures were detected between forest and the two unforested LUTs, arable land and grassland, revealed by the finding that *M. robertsii* and *M. guizhouense* were not detected in forest and *M. brunneum* was represented by four forest-specific genotypes. Although population structures did not differ between the two unforested LUTs, *Metarhizium* spp. abundance was significantly higher in grassland compared to arable land.

### 4.1. M. brunneum Populations in Forested vs. Unforested LUTs

The forest sites of the NABO network provided characteristic soil environmental conditions, with reduced exposure to sunlight, low soil bulk density, and high basal respiration, as well as high carbon content and C:N ratio (Appendix A). These physical, chemical, and biological factors were correlated with *M. brunneum* population variation among the 10 forest sites and 20 arable land or grassland sites, indicating their importance to the structure of *Metarhizium* populations. The difference between *M. brunneum* populations in forested and unforested sites was explained particularly by the soil factor C:N ratio, which was also the only factor that significantly explained *M. brunneum* population variation within one LUT, i.e., among grassland sites. The C:N ratio also differed significantly between low and medium *Metarhizium* abundance groups among all LUTs and among forest sites (Table 1). Our results suggest that *Metarhizium* abundance and species and genotype diversity tend to be higher when the C:N ratio is medium–low (Table 1, Table 2 and Table 3). In contrast to our results, Uzman et al. [43] reported enhanced *Metarhizium* spp. abundance with increasing C:N ratios in vineyard soils, and Clifton et al. [22] showed a negative correlation between *Metarhizium* spp. abundance and N concentration, a factor indirectly related to C:N ratios in soils of conventional and organically farmed fields. The latter authors hypothesized that this effect may be due to a stimulation of certain microorganisms that exploit elevated N concentrations and, subsequently, may outcompete *Metarhizium* spp. In vitro studies have demonstrated that the proportion of C and N in culture media considerably affects the production of biomass, conidia, and insecticidal molecules as well as the virulence of *Metarhizium* spp. [65,66,67]. These laboratory experiments were performed with a variety of C and N sources in artificial media, and, therefore, it remains unresolved as to what extent these findings may apply to soil.

### 4.2. Metarhizium Populations in Arable Land vs. Grassland

Although *M. brunneum* and *M. robertsii* population structures in arable land and grassland were not significantly different, we detected significant differences between the two LUTs in overall abundance as well as the abundance group associations of the sites. Mean CFU g^−1^ values in the grassland sites were approximately twice as high as the values in arable land. In addition, in grassland, *Metarhizium* was present with high or medium abundance at all the sites, whereas in arable land, *Metarhizium* was present with medium and low abundances and not even detected at two sites. Interestingly, at the forest sites, the fungus was detected at only 4 out of 10 sites, all belonging to the medium abundance group. The mean abundance was similar to the one detected in arable land when considering only sites where the fungus was detected. MAP is a factor that characterizes the different sites in terms of precipitation. It has been shown to correlate with soil humidity [68], which is well recognized as a critical factor for fungal abundance and the colonization of soil (e.g., [69]). The fact that the MAP values in our study were significantly higher at sites with medium abundance compared to sites with low *Metarhizium* abundance indicates that the MAP in our system also affected soil humidity, which, in turn, induced *Metarhizium* growth. However, we did not find any within the LUT correlation of MAP with *Metarhizium* abundance. Other factors that were significantly correlated with *Metarhizium* abundance groups within different LUTs, e.g., altitude, clay, or sand, did not reveal significant differences among LUTs, which may indicate that these factors characterize geographical differences among the sites rather than LUTs.

Direct comparisons of different studies are generally difficult to perform due to differences in isolation techniques, sampling schemes, habitat definitions, geographic and/or climatic zones. Additional local factors such as climat, vegetation, and geographic history may increase the complexity of interacting factors. However, in accordance with our results, Keller et al. [27] also reported a higher abundance of *Metarhizium* spp. (*M. anisopliae* complex) in meadows (semi-managed unforested land) compared to agricultural land in Switzerland. Furthermore, Schneider et al. [28] detected higher *Metarhizium* clade1 (*M. anisopliae* species complex) ITS copy numbers using real-time PCR in permanent grasslands and improved field margins compared to arable land. Such unforested land types do not receive pesticide treatments and are not subject to soil disturbances such as tillage. A number of studies have shown that soil receiving reduced or no soil management, including organically farmed soils, will allow a higher abundance of *Metarhizium* or entomopathogenic fungi in general [22,29,30,70,71]. Interestingly, it has been shown that, in contrast, tillage may also increase *Metarhizium* spp. abundance [20,21]. The authors of these studies have argued that tillage may disperse *Metarhizium* propagules, resulting in a more homogenous distribution of the fungus in the soil and, thus, increasing its overall abundance.

### 4.3. Association Metarhizium Species and Genotypes to LUTs

We observed different affinities of *M. brunneum*, *M. robertsii*, and *M. guizhouense* to the three LUTs. While *M. brunneum* was detected in all LUTs (73.3% of the sites) and was the dominant species at 16 out of 22 sites as well as in the three abundance groups, *M. robertsii* and *M. guizhouense* were detected in arable land and grassland only (36.6% and 10.0% of the sites). However, whereas *M. robetsii* was detected within low and medium abundance groups in arable land and low and high abundance groups in grassland, *M. guizhouense* was detected only in the medium abundance group in arable land and medium and high abundance groups in grassland. The *M. brunneum* genotypes detected at the forest sites were specific to forests and clustered separately from the genotypes detected at the arable land or grassland sites. In addition, certain MLGs were particularly dominant and present at multiple sites across the country (MLG2, MLG13, MLG 15, MLG 19). Species affinity to particular sites, regions, or habitats and the dominance of particular genotypes have been reported previously on a field level [21,36] as well as on a regional level [38,72,73]. Plant species composition characteristically differs between habitats or LUTs such as agricultural land, grassland, and forest. Previous studies have provided evidence that *Metarhizium* spp. can form stable associations with certain plant species [32,74], and that crop type may affect *Metarhizium* spp. diversity and abundance [21,31]. For instance, Cabrera-Mora et al. [31] predominantly isolated *M. brunneum* from bean-cultivated soil and *M. robertsii* from maize-cultivated soil, and Kepler et al. [21] detected higher *Metarhizium* abundance in soybean fields versus cornfields. Furthermore, it is well documented that arthropod populations vary substantially among LUTs [75]. Species associated with forests, for instance, bark beetles (*Scolytus* spp.), may not be present in unforested habitats, and pest species like wireworm spp. or western corn rootworm are linked to certain crop plants. The population structure of arthropod pathogenic fungi such as *Metarhizium*, which includes species and genotypes with diverse host ranges (from narrow to broad), is likely affected and shaped by arthropod populations (host selection). Furthermore, it has been observed that small soil arthropods, such as collembolans and mites, may be involved in the dispersal of fungal propagules in the soil, which may also have an effect on fungal population development [76]. The data reported here and observations made by others strongly indicate the importance of plant and arthropod populations in particular habitats and LUTs as the driving force for shaping the population structure of entomopathogenic fungi such as *Metarhizium* spp.

## 5. Conclusions

The 30 sites investigated in this study represent 3 well-defined and characterized habitat types in Switzerland. Being part of the Swiss soil-monitoring network (NABO), the sites have been cultivated, maintained, and monitored for the last three decades according to the same procedures and protocols, and therefore, represent well-established habitats. This long-term experimental system allowed us to unravel the significant impacts of the different LUTs on *Metarhizium* abundance, species diversity, and their population structures and indicated the complexity of environmental factors involved. With the detection of LUT-specific MLGs of *M. brunneum* and the absence of *M. robertsii* and *M. guizhouense* at forest sites, the study has emphasized the significant impact of forested versus unforested LUTs on *Metarhizium* population structures. These results will provide an important base to further explore other factors that may shape *Metarhizium* populations, which, for instance, may include the effects of plant populations as well as the diversity of arthropod populations present in the soil. Information compiled from such system approaches will reinforce the criteria for the selection of LUT-adapted fungal strains for evaluation as biological control agents. Future biological control strategies may not only include strains that are most virulent to particular pest insects but also those adapted to specific habitats, crops, or even sites. Such strategies may also include the manipulation of LUTs, for instance, by the cultivation of plant species that interact with these beneficial fungal species and support their growth, even in the occasional absence of a particular pest.

## Figures and Tables

**Figure 1 microorganisms-09-01380-f001:**
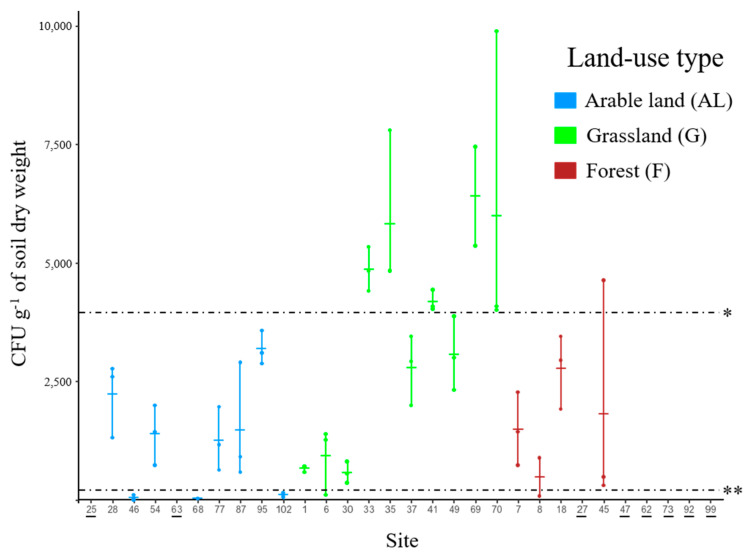
*Metarhizium* spp. abundance represented as dot plots of colony-forming units (CFU g^−1^ of soil dry weight) for different sites (30 NABObio sites). Dots represent values of CFU g^−1^ of soil dry weight in each sample per site, and the intersection represents the mean of CFU g^−1^ of soil dry weight per site. * Dashed line (4000 CFU g^−1^ of soil dry weight) separates the high abundance group from the medium abundance group; ** Dashed line (150 CFU g^−1^ of soil dry weight) separates the medium abundance group from the low abundance group) (Appendix A). Site numbers where *Metarhizium* was not detected are underlined.

**Figure 2 microorganisms-09-01380-f002:**
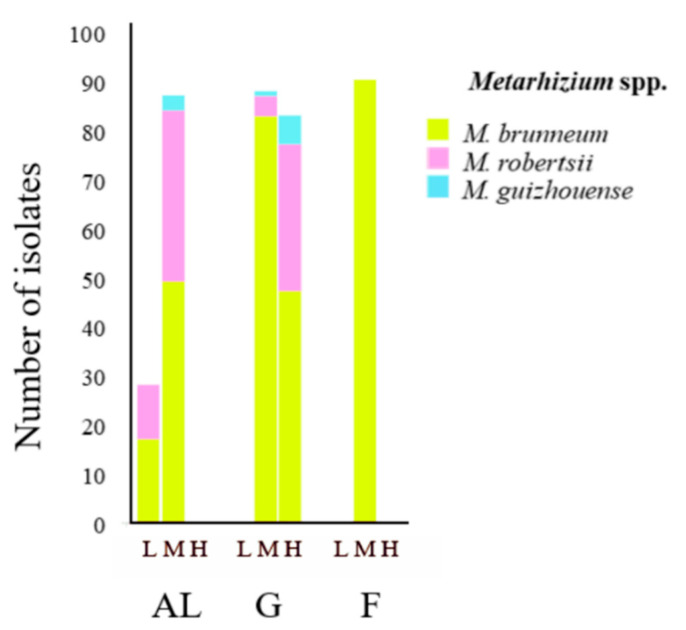
Number of isolates recovered for the three species identified (*M. brunneum*, *M. robertsii*, and *M. guizhouense*) in each of the three land-use types—arable land (AL), grassland (G), and forest (F)—separated according to the three abundance groups, low (L), medium (M), and high (H) (see Figure 1).

**Figure 3 microorganisms-09-01380-f003:**
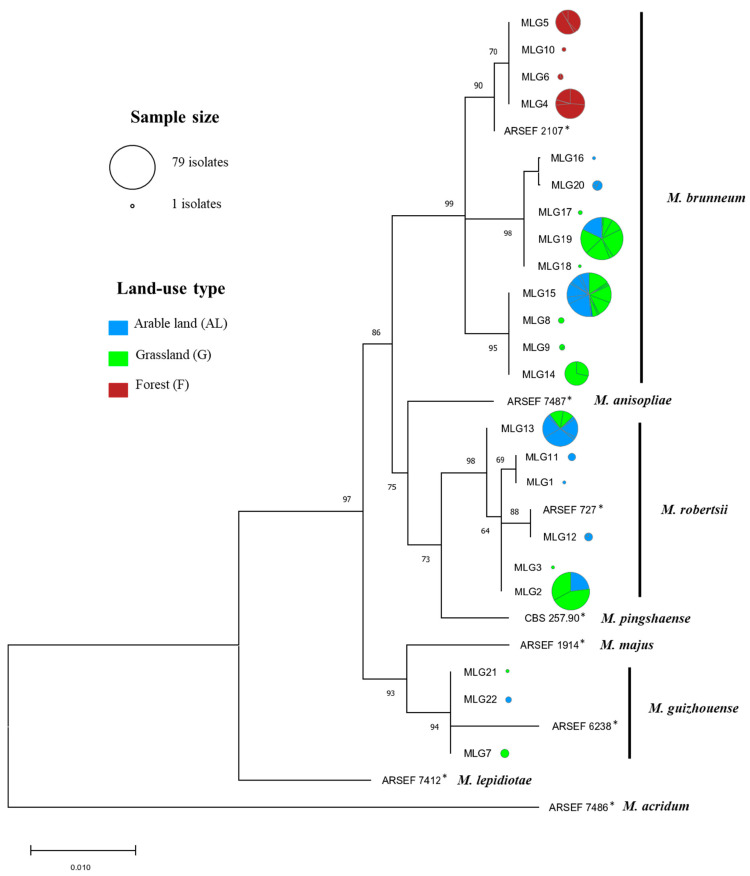
Maximum likelihood phylogenetic tree based on the alignment of 5′-TEF-1α sequences representing each of the 22 different multilocus genotypes (MLG). Bootstrap values > 60%, calculated with 1000 replicates, are shown. The bar scale indicates 0.01 changes per nucleotide. Each circle represents a unique MLG, and circle size indicates the number of isolates recovered. Colors within the circles represent the fraction of clones obtained per land-use type, and vertex size represents the fraction of clones isolated at sites where the MLG was detected.

**Figure 4 microorganisms-09-01380-f004:**
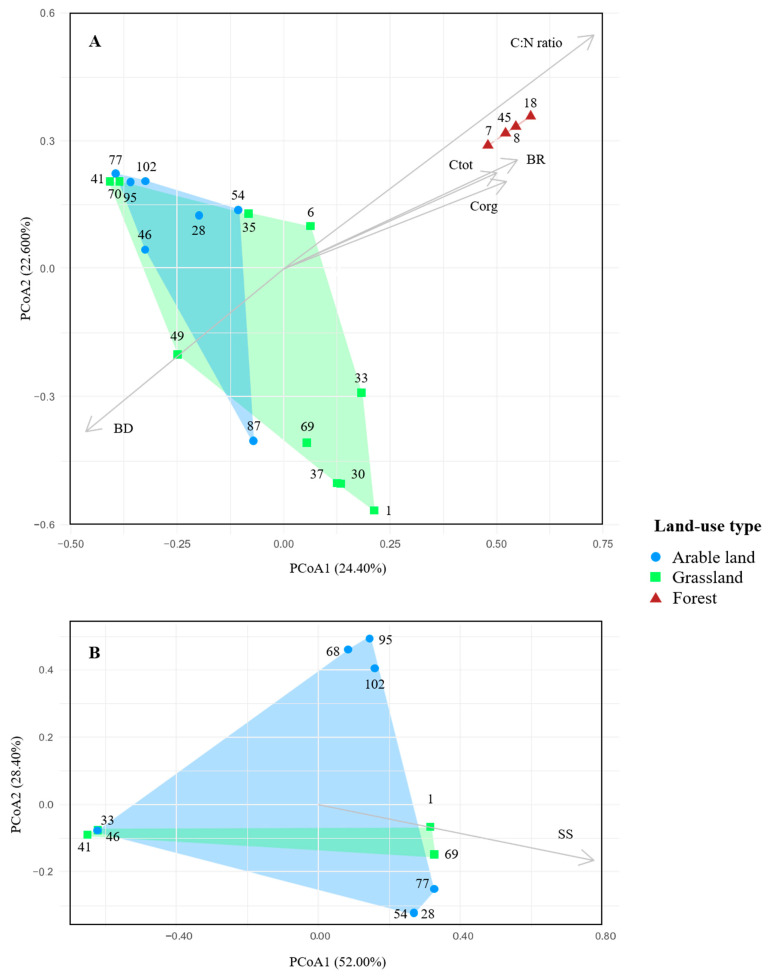
Principal coordinate analysis (PCoA) based on a Bray–Curtis dissimilarity matrix showing differences between (**A**) *M. brunneum* and (**B**) *M. robertsii* populations at the sites of three land-use types: arable land, grassland, and forest. Circles (arable land), squares (grassland), and triangles (forest) represent *M. brunneum* or *M. robertsii* populations at the different sites (site numbers indicated). Convex hulls enclose samples of the same land-use type. The percentage of variation explained by the plotted principal coordinates is indicated on the axes. Vectors represent the best fitted and explained environmental factors, and vector lengths represent the strength of correlation. C:N ratio: ratio of organic carbon and total nitrogen; Ctot: total carbon; Corg: organic carbon; BR: basal respiration; BD: bulk density; SS: soil skeleton.

**Table 1 microorganisms-09-01380-t001:** Summary of soil and environmental factors that differ significantly among *Metarhizium* spp. abundance groups (high, medium, and low) and between and within land-use types (LUTs).

LUT	Factor ^(1)^	High				Medium				Low				ANOVA		
		Mean	SD	Minimum	Maximum	Mean	SD	Minimum	Maximum	Mean	SD	Minimum	Maximum	F-Value	*p*-Value	Pattern ^(2)^
All LUT	C:N ratio	9.6	0.9	8.9	10.0	11.1	3.1	8.0	17.7	14.2	5.5	8.6	27.0	8.72	0.0004	H = M ≠ L
	MAP [mm]	1496.2	321.0	1090.0	1910.0	1342.4	363.2	962.0	2140.0	1116.7	368.5	528.0	1838.0	6.77	0.0019	H = M ≠ L
Arable land	Altitude [masl]	- ^(3)^	-	-	-	549.4	167.9	336.0	830.0	449.6	55.6	379.0	545.0	4.78	0.0374	M ≠ L
	Clay [% sdw]	-	-	-	-	17.6	4.6	11.5	23.8	32.5	19.1	5.8	59.0	8.69	0.0064	M ≠ L
	Sand [% sdw]	-	-	-	-	42.2	9.8	31.0	54.0	22.4	11.4	11.0	36.1	26.15	<0.0001	M ≠ L
	Soil skeleton [% sv]	-	-	-	-	4.0	0.8	3.1	4.9	0.7	0.8	0.0	2.0	135.85	<0.0001	M ≠ L
	Total Carbon [% sdw]	-	-	-	-	2.0	0.7	1.1	3.2	2.8	0.9	1.8	4.3	8.55	0.0068	M ≠ L
	DNA [µg/mg sdw]	-	-	-	-	25.5	9.1	14.0	45.0	17.9	3.8	13.0	26.0	8.85	0.006	M ≠ L
Grassland	Silt [% sdw]	42.3	8.4	34.3	55.0	35.6	7.9	27.0	49.9		-	-	-	5.10	0.0319	H ≠ M
Forest	Altitude [masl]	-	-	-	-	745.3	269.9	525.00	1180.0	1115.8	408.7	505.0	1655.0	7.60	0.0101	M ≠ L
	Clay [% sdw]	-	-	-	-	31.1	9.1	18.8	42.0	17.8	8.3	7.0	30.5	17.30	0.0003	M ≠ L
	Sand [% sdw]	-	-	-	-	31.1	8.7	18.0	39.0	47.2	21.0	17.5	71.0	6.35	0.0177	M ≠ L
	C:N ratio	-	-	-	-	15.5	2.2	12.0	17.7	18.2	4.3	13.7	27.0	4.25	0.0488	M ≠ L

^(1)^ Environmental factors that revealed significant differences are shown. For the complete analysis of the 15 environmental factors evaluated, see Appendix A. MAP: mean annual precipitation (1981 to 2015); masl: meters above sea level; sdw: soil dry weight; sv: soil volume. ^(^^2)^ Significant differences in pairwise tests between the proportion of colony-forming units (CFU) (*p* < 0.05, Tukey-HSD test). ^(3)^ *Metarhizium* was not detected at this abundance level.

**Table 2 microorganisms-09-01380-t002:** Numbers of isolates, multilocus genotypes, and genotype diversity among the three *Metarhizium* species—*M. brunneum*, *M. robertsii*, and *M. guizhouense*—for three different LUTs: arable land, grassland, and forest.

	*Metarhizium* spp.		*M. brunneum*				*M. robertsii*				*M. guizhouense*			
Land-Use Type	N	MLG	N	MLG	H	E.5	N	MLG	H	E.5	N	MLG	H	E.5
Arable land	115	10	66	4	0.99	0.75	46	5	1.09	0.60	3	1	-	-
Grassland	171	12	130	7	1.33	0.75	34	3	0.70	0.74	7	2	-	-
Forest	63	4	63	4	0.96	0.80	-	-			-	-	-	-
Total	349	22	259	13	1.84	0.72	80	6	1.17	0.74	10	3	-	-

N: total number of isolates; MLG: number of unique multilocus genotypes; H: Shannon–Wiener index; E.5: evenness.

**Table 3 microorganisms-09-01380-t003:** Overall permutational multivariate analysis of variance (PERMANOVA) for individual soil and environmental factors, which significantly affect *M. brunneum* and *M. robertsii* population structures among and within each land-use type. Analyses were based on Bray–Curtis distances.

	*M. brunneum*					*M. robertsii*		
	Among 3 LUTs	Grassland– Arable Land	Arable Land	Grassland	Forest	Between 2 LUTs	Arable Land	Grassland
Variable	*p*-Value (R^2^)	*p*-Value (R^2^)	*p*-Value (R^2^)	*p*-Value (R^2^)	*p*-Value (R^2^)	*p*-Value (R^2^)	*p*-Value (R^2^)	*p*-Value (R^2^)
Land-use type	**0.0001 (37%)**	0.0513	-	**-**	-	0.4665	-	-
C:N ratio	**0.0001 (25%)**	0.1149	0.9821	**0.0074 (26%)**	0.2083	0.5366	0.4599	0.8333
Basal respiration	**0.0071 (13%)**	0.0928	0.3016	0.7186	0.1667	0.1327	0.4595	0.6668
Organic Carbon	**0.0039 (12%)**	0.0861	0.5563	0.6421	0.2083	0.2562	0.6476	0.8333
Total Carbon	**0.0055 (11%)**	0.1657	0.5891	0.6488	0.2083	0.1403	0.3230	0.8333
Bulk density	**0.0055 (13%)**	0.2375	0.5190	0.0893	0.2500	0.1333	0.6103	0.1667
Soil skeleton	0.5821	0.4503	0.8032	0.5633	0.8333	**0.0098 (33%)**	0.3008	0.3333

R^2^ values are shown in parentheses when significant (in bold). For the complete analysis, see Appendix A.

## Data Availability

All additional data can be obtained from the corresponding author upon reasonable request.

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
