# Peer review of "Land-Use Type Drives Soil Population Structures of the Entomopathogenic Fungal Genus Metarhizium"

_microorganisms, 2021, doi:10.3390/microorganisms9071380_

Round 1
Reviewer 1 Report
See word file below.

Author Response
Response to Comments of Reviewer 1
Review Microorganisms Fernandez-Bravo et al.
Line numbers provided in the responses refer to the revised manuscript.
Point 1. 20. I did not find the MLG acronym defined in the test of the manuscript, just here in the abstract.
Response 1: The acronym definition for MLG has been added in line 201.
Point 2. 48. I have trouble with this statement that the “fact of a lack of dependance on insects ” should have any specific relationship with use of the fungus as a pest control agent. I don’t understand the link that the authors intend by this statement. Natural or endemic fungal density and/or diversity is not likely to be related to important considerations when developing EPF (entomopathogenic fungi) as biocontrol agents. Additional relationships may need to be identified or more clearly articulated.
Response 2: We agree with the reviewer. This statement was confusing. We have rephrased the text to provide a concise and meaningful statement. Lines 49 to 52.
Point 3. 125. Consider changing the sentence structure, “The Swiss soil-monitoring network established in 1984, monitors 103 sites for physical, chemical and biological parameters.”
Response 3: The sentence has been modified. Lines 129-132.
Point 4. 144. I question the MAP data. It seems odd to record precipitation for the sites after the samples were collected, especially when samples were collected in the first half of 2016. It would be more reasonable to me for MAP data for a site to include the year preceding the date of each sample. Precipitation after sampling seem irrelevant as a contribution to fungal density or diversity.
Response 4: Mean annual precipitation is a mean calculated for the years 1981 to 2015. This value characterizes moisture conditions of the sites rather than the immediate amount of precipitation received in a particular time of the year. Based on the comment we have realized that our description of the value was not correct. We have improved the description in the Material and Methods section. Lines 150 to 151.
Accordingly, we have rephrased the discussion on this topic in lines 442 to 452.
Please also refer to point 8 of reviewer 3.
Point 5. 154. As a user of selective media, I would appreciate an indication of the selective agent used in the plates without needing to search the reference cited.
Response 5: We now indicate the selective compound in line 161-162.
Point 6. 179. (and throughout the manuscript) When reporting companies in the USA, please include the state as many states have cities with the same name.
Response 6: Done.
Point 7. 215. Were the CFU data transformed for analysis? It seems appropriate to use a log transformation for analysis of CFU data? It is appropriate to use log (x+1) transformation to comply with the assumption of normality in the data for ANOVA and it may help identify significant differences among samples.
Response 7: We have tested the residuals of untransformed data and they comply with the assumption of normality (Shapiro-Wilk test among LUTs: W=0.966; P(W)>0.1; and among sites: W=0.812; P(W)>0.05) . Therefore, the CFU data were not transformed for the analyses.
However, we also performed the analyses with log (x+1) transformed data and the significance and pairwise comparisons did not differ from the ones shown in the manuscript. The significance values obtained after transformation were F2,19=4.31; p-value=0.0287 for differences among LUTs and F29,58 = 65.03; p-value < 0.0001 for the differences among sites.
Point 8. 327 (227). Consider text edit “Using selective media for culture, the Metarhizium genus…”
Response 8: The sentence has been modified. Line 248.
Point 9. 244. I though the zero sites were omitted from this analysis, but the F degrees of freedom (29,87) suggest they were included here. Am I wrong?
Response 9: You are correct. Comparisons of the LUTs did not include the zero sites (as described in the Materials and Methods section). Therefore, this analysis included only 22 sites with a degree of freedom of 2 (line 250). For the comparison among sites, all sites were included with a degree of freedom of 29 (line 255). Thank you for this comment; we have realized that we did not provide the correct values for the degree of freedom of the ANOVA residuals. Corrections have been made in the new version of the manuscript (df of ANOVA residuals 19 and 58 for the two analyses, 58 because at two sites we only have data for two soil samples). In addition, we have checked again all the statistical values throughout the manuscript.
Point 10. 247. What criteria were used to define these groups? The 150n and 4000 breaks seem arbitrary. I would use a more logarithmic break like 200 and 2000. I think this might give you more even distribution of your sites, although it may complicate your results and discussion.
Response 10: The groups were defined based on the abundance data obtained and our goal was to test whether any of the assessed environmental factors may explain the observed groups. For this reason, we did not select a mathematically based group definition.
Point 11. 259. Did you consider presenting your figure using a log scale for the Y axis? Also, I have trouble discerning the gray scale for your data in my B/W printed copy in this and other figures.
Response 11: We also plotted the data with a log scale for the Y-axis, when performing all the analyses. However, the data presented in log scale are more difficult to read and most importantly, the three abundance groups discussed in Point 10 are not as distinct as in the presentation in linear scale. The disadvantage of this representation is that low abundance values are less resolved. For this reason, we mention the sites, where Metarhizium was not detected in the text (lines 248-250), and in addition, we now clearly indicate these sites in Figure 1.
Point 12. 266. consider “measured” or “recorded” to replace “determined”. Also, This section made it obvious to me that the authors did not indicate the relationship of the significant factors on the fungal density or diversity, just that they were different. “Low C:N ratios were reflected as ???? abundance of fungus”? I kept looking for the direction of the relationships.
Response 12: Done. Line 277.
In addition, we have change the tittle of section 3.2. to “Effect of environmental factor on Metarhizium abundance groups” to point out the focus of the paragraph. Line 276.
We provide an interpretation of the data in the discussion. Lines 408-431. Please also see point 15 below.
Point 13. Table 1. What are the units for the various factors?
Response 13: We have added the units in Table 1.
Point 14. 283. For accuracy consider, “…isolates were collected from 22 of 30 sites sampled…”
Response 14: The text has been modified to improve clarity. Lines 297-298.
Point 15. 405. This is the kind of relationship I was looking for in the results section. (see comment for line 266)
Response 15: Done. Please also see response to point 12.
Point 16. 414. Philosophically, Could the adaptation to infect insects be a relatively new one for a poor N nutrient competitor to take advantage of a resource (insects) this is not available to otherwise highly competitive but non pathogenic fungi? I don’t expect a response to this comment.
Response 16: We agree, based on currently available data this is a valid hypothesis worth to be tested.
Point 17. 501. Consider editing to read “… LUT adapted fungal strains for evaluation as biological control agents”.
Response 17: The sentences have been rephrased. See also point 11 of Reviewer 3. Line 520-524.
Point 18. 561. Capitalize “Helicoverpa and Metarhizium”.
Response 18: Done. Line 586.
Reviewer 2 Report
This paper presents a detailed analysis of the relationship between LUTs and the distribution of fungi of the genus Metarhizium.
All tasks set in the study were completed. The methodological part is in line with modern ideas.
In my opinion, the discussion lacks a comparison of the obtained data with similar studies in other countries, in particular, in the USA, Canada, etc.
For example, there are studies that show the distribution of the same species of fungi by soil type.
Author Response
Response to Comments of Reviewer 2
Point
In my opinion, the discussion lacks a comparison of the obtained data with similar studies in other countries, in particular, in the USA, Canada, etc.
For example, there are studies that show the distribution of the same species of fungi by soil type.
Response:
We know a number of studies performed in different countries that assess presence, abundance and diversity of Metarhizium spp. and we have introduced several of them in the introduction. However, as we indicated in the discussion it is very difficult or even impossible to compare the different studies because the LTUs or habitats investigated differ among studies and often particular LTUs exist only in a certain region due to e.g. the climatic conditions and/or the plants cultivated.
The goal of our study was to specifically compare the three dominant land-use-types in Switzerland and to investigate habitat selectivity and the factors that may drive population structures and abundance of Metarhizium spp. in these land-use-types. In our discussion we have considered aspects that allowed a comparison with other studies, e.g. the effect of C:N ratio. In order to avoid being too speculative, we would like to keep the discussion at this level of comparisons.
Reviewer 3 Report
Dear authors,
I would like to congratulate you on this tidy and complete original research work. The manuscript is well written, the methods well described and the results supported by appropriate statistical tests. My major concern is about the conclusion where I suggest some improvements.
In this regard, I have only a few comments and minors corrections for you to improve the final article.
Abstract
Line 15 & 20. Eliminate the short names for the abstract
Line 20. Typo ". With"
Materials and Methods
Line 158 - 159. The method applied to obtain monosporic cultures is not clear. Please, refer to a cite or describe briefly the methodology used.
Results
Line 268. Please, instead "mean annual precipitation (MAP)"
Table and Figures
In Tables and Figures, when the species of Metarhizium or even the genus is mentioned, these are not written in italics. Since you have written them correctly in all the manuscript, I supposed that could be an edition issue. Please check.
Table 1. Add. "abundance groups (high, medium, and low)"...
In table one, there are no data in the high Metarhizium abundance group for Arable land and Forest LUTs. It would be helpful if you can specify in the table if these missing data were not determined/observed (ND/NO)...
Lines 431 - 436. Rephrase. Which factor, MAP? More important, be clear about the effect you are trying to explain. I found these lines a bit confusing.
Line 485. Substitution. "shaping the population structure"...
Line 489. Instead, use the short name "NABO"
Conclusions
Lines 497 - 499. Rephrase as a conclusion. How your results are founding or contributing to further studies?
Lines 499 - 501. Please exemplify or mention at least one new biological control strategy that could be developed based on the suggested studies
Author Response
Response to Comments of Reviewer 3
Line numbers provided in the responses refer to the revised manuscript
Abstract
Point 1. Line 15 & 20. Eliminate the short names for the abstract
Response 1: Done. Lines 19 and 23.
Point 2. Line 20. Typo ". With"
Response 2: We have checked the indicated position in the text but we did not find a typographic mistake.
Materials and Methods
Point 3. Line 158 - 159. The method applied to obtain monosporic cultures is not clear. Please, refer to a cite or describe briefly the methodology used.
Response 3: We have added a brief description of the procedure. Lines 166-169.
Results
Point 4. Line 268. Please, instead "mean annual precipitation (MAP)"
Response 4: Done. Line 279-280.
Table and Figures
Point 5. In Tables and Figures, when the species of Metarhizium or even the genus is mentioned, these are not written in italics. Since you have written them correctly in all the manuscript, I supposed that could be an edition issue. Please check.
Response 5: Done. All the figures and tables have been verified.
Point 6. Table 1. Add. "abundance groups (high, medium, and low)"...
Response 6: Done. Line 289.
Point 7. In table one, there are no data in the high Metarhizium abundance group for Arable land and Forest LUTs. It would be helpful if you can specify in the table if these missing data were not determined/observed (ND/NO)...
Response 7: We have added a footnote to explain the meaning of “-“. Line 294.
Point 8. Lines 431 - 436. Rephrase. Which factor, MAP? More important, be clear about the effect you are trying to explain. I found these lines a bit confusing.
Response 8: We have rephrased the sentences to improve our argumentation and we have included an additional reference to support the arguments. Lines 442-451.
Please also refer to point 4 of reviewer 1
Point 9. Line 485. Substitution. "shaping the population structure"...
Response 9: Done. Line 518.
Point 10. Line 489. Instead, use the short name "NABO"
Response 10: As part of the conclusions, we believe providing the full name would be helpful to the reader. Therefore, we would like to keep the original wording.
Conclusions
Point 11. Lines 497 - 499. Rephrase as a conclusion. How your results are founding or contributing to further studies?
Response 11: The sentences have been rephrased. Line 520-524.
Point 12. Lines 499 - 501. Please exemplify or mention at least one new biological control strategy that could be developed based on the suggested studies
Response 12: We have elaborated on the statements and provide more specific arguments. Lines 524-527.
Reviewer 4 Report
This is a comprehensive and well executed study on the relationships of land use types and species of an entomopathogenic fungus. The uniqueness of this study is that it has been able to take advantage of the Swiss soil monitoring network which provides much background information on the soils used. Of course there is still much more to learn, however this study provides an important part of the puzzle.
I have only very minor points for the authors to consider:
51-55 Awkward sentence. Please reword.
MAP should be defined in table 1
Species names should be in italics in the table and figure headings
When presenting percentages, is it really necessary to provide two decimal places. Would it not suffice to round off to one decimal place it not even to whole numbers? How significant are a .01 percentage in these results?
391 abundance is a singular noun: abundance was or abundances were
395 missing a comma
Author Response
Response to Comments of Reviewer 4
Line numbers provided in the responses refer to the revised manuscript
Point 1. 51-55 Awkward sentence. Please reword.
Response 1: We have rephrased the sentences. Line 53-60.
Point 2. MAP should be defined in table 1
Response 2: We have added a definition of MAP in the footnote of Table 1 and we have improved the description in the Materials and Methods section. Line 150-151 and 279-280.
Point 3. Species names should be in italics in the table and figure headings
Response 3: We have verified and corrected the format for species names throughout the figures and tables as well as the main text.
Point 4. When presenting percentages, is it really necessary to provide two decimal places. Would it not suffice to round off to one decimal place it not even to whole numbers? How significant are a .01 percentage in these results?
Response 4: Done. We now provide all the values with one decimal only, except for the F and p values.
Point 5. 391 abundance is a singular noun: abundance was or abundances were
Response 5: Done. Line 406.
Point 6. 395 missing a comma
Response 6: Done. line 410.